# Effects of Land Use Changes on Ecosystem Service Value in Xiangjiang River Basin, China

**Ze Zhou [1], Bin Quan [1,2,\*] and Zhiwei Deng [1]** 

1   College of Geography and Tourism, Hengyang Normal University, Hengyang 421002, China
2   HIST Hengyang Base, Hengyang Normal University, Hengyang 421002, China
\*   Correspondence: binquan@hynu.edu.cn; Tel.: +86-152-0059-2617

**Abstract:** The study of land use and land cover (LULC) change and the evolution of ecosystem service values (ESV) is important for optimizing the allocation of regional land resources and achieving sustainable development, but traditional land use analysis methods cannot dig deeper into the information on the internal transitions between different land types. Therefore, in this paper, we use the component analysis and Intensity Analysis methods to systematically analyze the land use landscape changes at multiple levels. In addition, the spatial and temporal dynamic characteristics of ESV in the Xiangjiang River Basin are carved with the help of equivalence factors and GIS spatial analysis methods, which reveal the response of ESV changes to land use changes in the Xiangjiang River Basin in the past four decades. Our results showed that, (1) in the past 40 years, the intensity of land use change in the Xiangjiang River Basin has been increasing, mainly in quantity and shift. The increase of built-up and bare land and the decrease of cultivated land are stable and active, and the loss of forest land is large, but it is dormant. The loss of cultivated and forested land flows mainly to built-up land. (2) In the Xiangjiang River Basin, ESV increases first and then decreases, mainly in the middle and high grades and changes faster in the east. (3) The cross-sensitivity coefficient reflected that the net conversion of cultivated land to forest land and water area had a promoting effect on ESV. Our results provide important knowledge to inform land use decisions and facilitate sustainable development in the Xiangjiang River Basin.

**Keywords:** LULC; ecosystem service value; Intensity Analysis; Xiangjiang River Basin

## 1. Introduction

Ecosystems are irreplaceable natural assets for humans in the process of survival and development, contributing directly or indirectly to human well-being through their ecological features, functions, processes, and products [1]. Its ecosystem service value (ESV) reflects the level to which the life-supporting goods and services they provide contribute to the needs of human life [2,3], and the main indicators include supply service, regulation service, support service, and aesthetic landscape service [4]. With the deepening global change research, a range of issues that impact on ecosystems and socioeconomic sustainability are widely concerned [5,6], in which researchers have found that strong changes in land use and land cover (LULC) are a major factor in the erosion and damage to ESV [7]. LULC influences the value of ecosystem services by affecting the patterns, processes, and functions of ecosystems [8,9]. However, since the reform and opening up in China, along with the rapid economic development and the accelerating urbanization process, human activities represented by urban and cultivated land expansion have led to the fragmentation and degradation of regional habitats and also had a profound impact on the regional ecosystem service level [10,11]. For example, the high intensity of human use of grasslands and forests not only causes degradation of ecosystems and reduction of biodiversity but also causes weathering of the land, thus weakening the value of the associated ecosystems, such as water conservation and soil protection [12]. Therefore, how

to scientifically allocate limited spatial resources and reasonably coordinate the relationship between environmental protection and economic development has become a hot issue of concern for policymakers and scholars related to the problem.

The accounting of ecosystem services as an objective value that permeates all stages of human social development is closely related to human well-being. Research on ecosystem services goes back to the 1970s when SCEP first introduced the concept of ecosystem service functions in the Report on Human Impact on the Global Environment [13]. After that, Daily elaborated on the evaluation of its value in Nature's Services Societal Dependence on Natural Ecosystems, published in 1997 [14]. In the same year, based on utility value and equilibrium value theory, Constanza et al. provided the initial assessment of the economic worth of global ecosystem services [15]. However, the composition, structure, and function of the ecosystem in different regions are diverse and different, which makes the intensity [16], mode [17], and effect [18] of LULC's influence on ESV vary. Therefore, the scholars Xie et al. modified the equivalence factor table according to the actual situation in China using the Willingness Survey value assessment method and combining expert knowledge, and it has been widely used [19,20]. In contrast, in early studies on the effects of land use change on ESV, researchers mostly considered the effects of a single land use type change on ESV [21,22], often ignoring the differences of ESV changes caused by transfer between different categories [23]. For example, the conversion of cropland to forest increases ESV, while the conversion of cropland to built-up land decreases ESV. In addition, land use change at different time scales has different characteristics, and the impact of its changes on ESV has shown great differences. However, the current research, which mostly focuses on the years after 2000, examines the effects of land use change on ESV on a somewhat small scale [24,25]. Therefore, mining the internal information of land use change on a long-term scale to clarify the impact of different types of land use transfer on ecosystem service value can further reveal the mechanism of land use change on the ecosystem service value.

Since the beginning of economic reforms in the late 1970s, China has seen unheard-of levels of economic growth and urbanization, which have also significantly altered the spatiotemporal structure of LULC [26]. However, LULC, as a form to represent the value of regional ecosystem services, its changes will inevitably affect the economic value of ESV. At present, the assessment of ESV is mainly concentrated in the developed region [27–29], and an in-depth quantitative analysis of ESV changes caused by long-term LULC changes in the medium-developed Xiangjiang River Basin is still lacking. Since the 1990s, the Xiangjiang River Basin, as the key area of Hunan's economic and urbanization development, has become a frequent area of natural disasters and the worst disaster area, which has seriously affected the sustainable development of the social economy [30,31]. In addition, a series of measures have been developed by local governments at all levels to ensure the ecological environment develops sustainably [32]. Therefore, it is necessary to evaluate the ecosystem service value of the Xiangjiang River Basin, which is the concentrated and leading area of agricultural modernization in Hunan Province.

In this article, we use the method of change composition and the Intensity Analysis to understand the intricacies of land use changes. By deeply mining the information of the transfer matrix, the change component analysis method can identify the variable components: quantity, exchange, and shift [33–35]. The Intensity Analysis is a quantitative mathematical analysis framework [36,37]. Compared with traditional land use quantitative methods, the Intensity Analysis has several advantages: First, it can systematically and quantitatively dig the information on land use change from the three levels of the interval level, category level, and transition level [38,39]. Second, it can more intuitively reflect the information on land use change [40]. At present, this method has achieved fruitful results in land use change [41,42], urban expansion [43], grassland desertification [44], and regional comparison [45,46]. However, in most of the LULC studies that have used the Intensity Analysis, there have been few studies on the economic benefits arising from LULC in conjunction with ESV. Therefore, we analyze the LULC of the Xiangjiang River Basin

from 1980 to 2020 and quantify the impact of LULC on the value of ecosystem services. This study aims to carry out the following objectives: (1) study of the total change, quantity, exchange, and shift change in land use change processes in the Xiangjiang River Basin over the past four decades; (2) explore the size, intensity, and characteristics of LULC change in the Xiangjiang River Basin; (3) to study the spatial and temporal situation of changes in the value of ecosystem services in the Xiangjiang River Basin over the past four decades; and (4) study of the impact of conversion of different categories on changes in the ESV. This study provides a reference for land and resources departments to optimize the allocation of land resources and coordinate resource utilization and sustainable ecological development.

## 2. Materials and Methods

### 2.1. Study Area

The Xiangjiang River is the second-largest tributary of the Yangtze River in China. It originates from the near peak ridge of Haiyang Mountain in Xing'an County, Guangxi, then enters Hunan Province. It runs through the mountains and basins in the transition from Nanling Mountain to Dongting Lake Plain. After receiving various tributaries along the way, it flows into Dongting Lake in two branches at Haohekou, Xiangyin County, with a total length of 856 km. The total drainage area is 96,253 km$^2$, between 111°01′ E~114°14′ E and 111°33′ N~113°22′ N. In this study, the whole Xiangjiang River Basin in Hunan Province is selected, covering 45 county-level administrative regions in Hunan Province, of which the trunk stream mileage in Hunan Province is 670 km, and the drainage area is 86,791 km$^2$, accounting for 78.27% of the total mileage and 90.2% of the total drainage area, respectively [47,48]. The Nanling Mountains and Luoxia Mountains in the basin are generally more than 1000 m above sea level, and the hills are more than 500 m below (Figure 1). They are typical subtropical humid monsoon climates. The Xiangjiang River Basin is a concentration and leading area for new industrialization, new urbanization, and agricultural modernization in Hunan Province. The industrial structure of the basin is dominated by secondary and tertiary industries, which account for 64% and 60% of the total output value of Hunan Province, respectively. Although the primary industry accounts for a relatively small share, it plays an essential supporting role in the economic development process of Hunan Province.

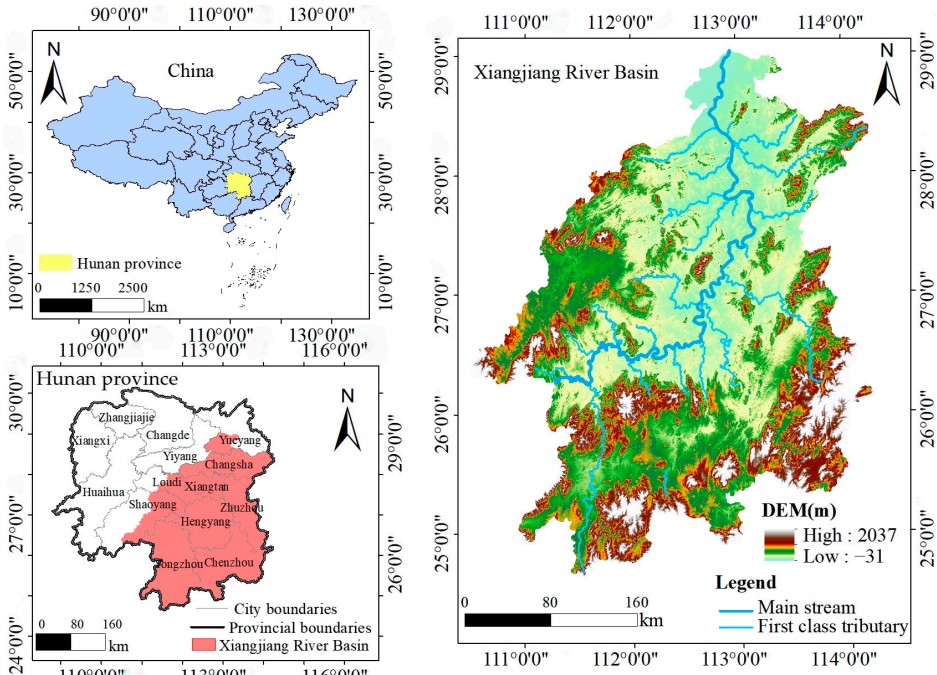

**Figure 1.** Location map of the study area.

*2.2. Data Sources*

The five periods of 30 m spatial resolution land use data of the Xiangjiang River Basin, all from the Resource and Environmental Science and Data Centre of the Chinese Academy of Sciences (http://www.resdc.cn/, accessed on 1 October 2021). It was reclassified by ArcGIS 10.8 and TerrSet 2020 to obtain cultivated, forest, grass, water, built-up, and bare land. The status and spatial and temporal distribution of LULC over the five periods are shown in Figure 2.

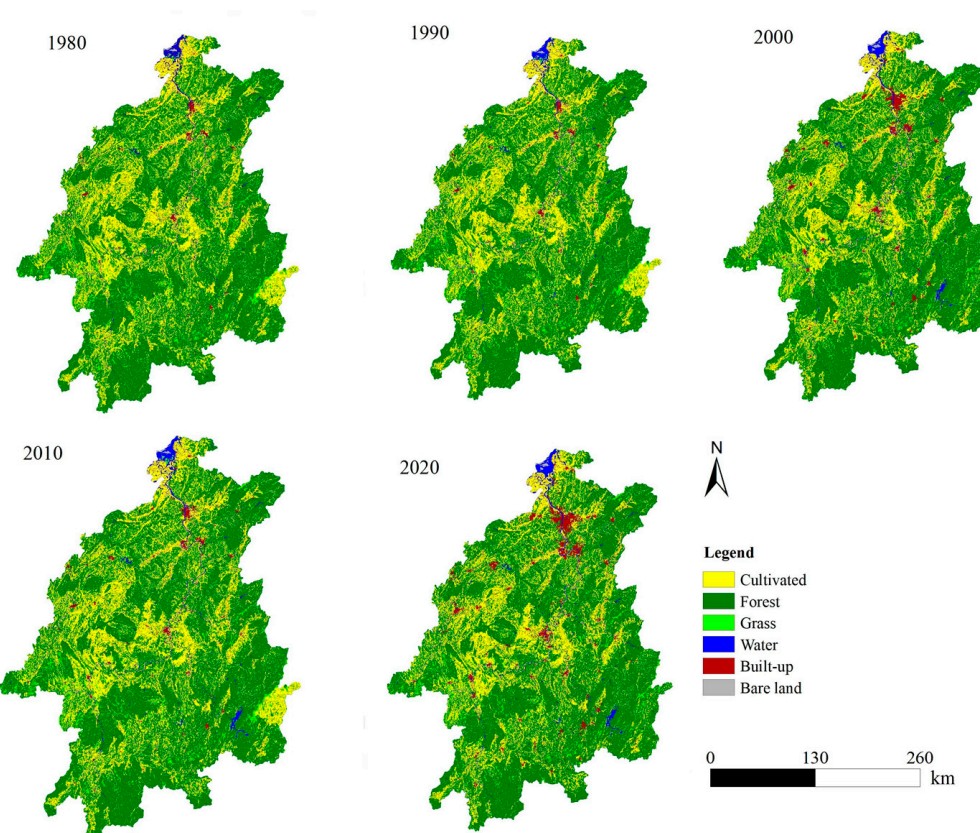

**Figure 2.** Five periods of land use status maps in the Xiangjiang River Basin.

The data of main grain market price and grain yield per unit area are from the National Food and Strategic Reserves Administration of China (http://www.lswz.gov.cn/html/zmhd/lysj/lsjg.shtml, accessed on 15 November 2021) and the National Bureau of Statistics of China (http://www.stats.gov.cn/, accessed on 15 November 2021), respectively. The value of built-up is not included because of its large negative benefits to the ecosystem.

*2.3. Research Methods*

2.3.1. Change Component Analysis

Change component analysis is to reveal the situation and way of change within the category, mainly including the total change, quantity, exchange, and shift change [33]. Equations (1)–(8) describe the calculation method and mathematical notation of the change component analysis (Table 1). The total change is calculated as Equations (1) and (2).

$$d_j = [\sum_{i=1}^{J} (C_{ij} + C_{ji})] - 2C_{jj} \tag{1}$$

$$D = \frac{\sum_{j=1}^{J} d_j}{2} = Q + E + M \tag{2}$$

**Table 1.** Mathematical notation of the Intensity Analysis and Change component analysis.

| Symbol | Cultivated |
|---|---|
| $T$ | Number of time points |
| $t$ | Index of the initial time point for a time interval |
| $J$ | Number of categories |
| $i$ | Index for a category |
| $j$ | Index for a category |
| $C_{tij}$ | Number of pixels that transition from category $i$ to category $j$ during interval $[Y_t, Y_{t+1}]$ |
| $C_{tji}$ | Number of pixels that transition from category $j$ to category $i$ during interval $[Y_t, Y_{t+1}]$ |
| $d_j$ | Annual difference for category $j$ during interval $[Y_t, Y_{t+1}]$ |
| $q_j$ | Annual quantity component for category j at interval $[Y_t, Y_{t+1}]$ |
| $e_j$ | Annual exchange component for category j at interval $[Y_t, Y_{t+1}]$ |
| $m_j$ | Annual shift component for category j at interval $[Y_t, Y_{t+1}]$ |
| $Q$ | Quantity of all categories at the time interval $[Y_t, Y_{t+1}]$ |
| $E$ | Exchange of all categories at the time interval $[Y_t, Y_{t+1}]$ |
| $M$ | Shift of all categories at the time interval $[Y_t, Y_{t+1}]$ |
| $D$ | Total change component of all categories at the time interval $[Y_t, Y_{t+1}]$ |
| $S_t$ | Annual change rate at interval $[Y_t, Y_{t+1}]$ |
| $U$ | Uniform change intensity at whole time interval |
| $L_{ti}$ | Intensity of loss in time interval $[Y_t, Y_{t+1}]$ for category $i$ |
| $G_{tj}$ | Intensity of gain in time interval $[Y_t, Y_{t+1}]$ for category $j$ |
| $T_{rin}$ | Intensity of transition from category $i$ to a special category $n$ at the interval $[Y_t, Y_{t+1}]$ |
| $W_{rn}$ | Uniform intensity from every category to category n at time $[Y_t, Y_{t+1}]$ |

Quantity change is caused by the net increase or decrease of the base area [33], as shown in Equations (3) and (4).

$$q_j = \left| \sum_{i=1}^{J}\left(C_{ij} - C_{ji}\right) \right| \tag{3}$$

$$Q = \frac{\sum_{j=1}^{J} q_j}{2} \tag{4}$$

Exchange change indicates that two categories have exchanged with each other. If, at time $T_1$, the land use type of a place A is category 1 and the land use type of a place B is category 2, and at time $T_2$, the land use type of place A becomes land category 2 and the land use type of a place B becomes land category 1, then category 1 and category 2 are exchanged two by two in spatial location [33], as shown in Equations (5) and (6).

$$e_j = 2\left\{ \left[ \sum_{j=1}^{J} MINIMUM(C_{ij}, C_{ji}) \right] - C_{jj} \right. \tag{5}$$

$$E = \sum_{j=1}^{J} e_j / 2 \tag{6}$$

Shift change indicates that three or more ground classes with equal area have moved in space, and the amount of displacement change is equal to how much these ground classes move in space [33]. The formula is as follows:

$$m_j = d_j - q_j - e_j \tag{7}$$

$$M = \sum_{j=1}^{J} m_j / 2 \tag{8}$$

### 2.3.2. Intensity Analysis

The Intensity Analysis is a mathematical framework that compares uniform intensities to observed intensities of temporal changes among categories [36]. It consists of a top-down composition of the interval, category, and transition levels. The interval level is used to explain whether LULC is fast or slow, and the category level further

compares a uniform intensity of change to observed intensities of loss and gain for each category during each time interval. The transition level provides a more in-depth analysis of whether the conversion of other categories to a special category is target or avoid. Equations (9)–(14) describe the calculation method and mathematical notation of the Intensity Analysis (Table 1).

(1) Interval level: Equations (9) and (10) calculate, in turn, the intensity of change for each time interval and the average intensity of change for the whole study period. If $S_t = U$, indicating that land use change is stationary; $S_t > U$ indicates that the land use change in this interval is fast; otherwise, it is slow [36].

$$S_t = \frac{\left\{\sum_{j=1}^{J}\left[\left(\sum_{i=1}^{J} C_{tij}\right) - C_{tjj}\right]\right\}/\mathrm{T}}{\sum_{j=1}^{J}\sum_{i=1}^{J} C_{tij}} \times 100\% \tag{9}$$

$$U = \frac{\sum_{t=1}^{T-1}\left\{\sum_{j=1}^{J}\left[\left(\sum_{i=1}^{J} C_{tij}\right) - C_{tjj}\right]\right\} / \sum_{t=1}^{T-1}\left[(Y_{t+1} - Y_t)\sum_{i=1}^{J}\sum_{j=1}^{J} C_{tij}\right]}{(Y_T - Y_1)} \times 100\% \tag{10}$$

(2) Category level: Equations (11) and (12) calculate the annual gain and loss intensity in the interval $[Y_t, Y_{t+1}]$, respectively. When $L_{ti} > S_t$ or $G_{tj} > S_t$, then the intensity of the decrease or increase category $j$ or category $i$ is active at that time; otherwise, it is dormant [36].

$$L_{ti} = \frac{[(\sum_{j=1}^{J} C_{tij}) - C_{tii}]/(Y_{t+1} - Y_t)}{\sum_{j=1}^{J} C_{tij}} \times 100\% \tag{11}$$

$$G_{ti} = \frac{[(\sum_{i=1}^{J} C_{tij}) - C_{tjj}]/(Y_{t+1} - Y_t)}{\sum_{i=1}^{J} C_{tij}} \times 100\% \tag{12}$$

(3) Transition level: Equations (13) and (14) calculate respectively the intensity of transition from other categories and equilibrium intensity of change to a special category [36]. If $T_{rin} > W_{rn}$, the gaining of category $n$ targets category $i$; otherwise, the gaining of category $n$ avoids category $i$ [36].

$$T_{rin} = \frac{C_{tin}/(Y_{t+1} - Y_t)}{\sum_{j=1}^{J} C_{tij}} \times 100\% \tag{13}$$

$$W_{rn} = \frac{\left[\left(\sum_{i=1}^{J} C_{rin}\right) - C_{rnn}\right]/(Y_{t+1} - Y_t)}{\sum_{j=1}^{J}[(\sum_{i=1}^{J} C_{tij}) - C_{tnj}]} \times 100\% \tag{14}$$

### 2.3.3. Estimating the Value of Ecosystem Services

We used the widely utilized equivalence factor method to calculate the value of ecosystem services in the Xiangjiang River Basin [15]. The Chinese ecosystem service coefficient correction method proposed by Xie et al. was used [19], whereby a standard economic value of ecosystem services equivalence coefficient is 1/7 of the economic value of food production per unit area of farmland. Using the production, sown area, and unit price of major food crops in Hunan Province from 1980 to 2020 as the base data, Equation (15) was applied to calculate the equivalent value of ecosystem services in the study area as 2724.57 CNY/hm². The value of ecosystem services per unit area was calculated for each land type in the Xiangjiang River Basin (Table 2).

$$VC_0 = \frac{1}{7}\sum_{i=1}^{n}\frac{m_i p_i q_i}{M} \tag{15}$$

where $VC_0$ is the value equivalent per unit area, CNY/hm²; $m_i$ is the unit yield of the $i$th food crop, kg/hm²; $p_i$ is the unit price of the $i$th food crop; CNY/kg; and $q_i$ is the sown area of the $i$th food crop, hm².

**Table 2.** Coefficient of ESV of different land use types in the Xiangjiang River Basin ($10^8$ CNY).

| Ecosystem Service Functions | | Cultivated | Forest | Grass | Water | Bare |
|---|---|---|---|---|---|---|
| Supply services | Food production | 2724.57 | 272.46 | 817.37 | 272.46 | 27.25 |
| | Raw material production | 272.46 | 7083.88 | 136.23 | 27.25 | 0 |
| Regulate service | Gas regulation | 1362.29 | 9536.00 | 2179.66 | 0 | 0 |
| | Climate regulation | 2424.87 | 7356.34 | 2452.11 | 1253.30 | 0 |
| | Hydrological regulation | 1634.74 | 8718.62 | 2179.66 | 55,526.74 | 81.74 |
| | Waste disposal | 4468.29 | 3569.19 | 3569.19 | 49,532.68 | 27.25 |
| Support services | Soil conservation | 3977.87 | 10,625.82 | 5312.91 | 27.25 | 54.49 |
| | Maintaining biodiversity | 1934.44 | 8882.10 | 2969.78 | 6784.18 | 926.35 |
| Cultural services | Aesthetic Landscape | 27.25 | 3487.45 | 108.98 | 12,069.85 | 27.25 |

The ESV is calculated as follows:

$$ESV = \sum A_i \times VC_i \tag{16}$$

$$ESV_f = \sum A_{if} \times VC_{if} \tag{17}$$

where $ESV$ and $ESV_f$ refer to the entire ecosystem and the value of individual ecosystem services, $A_i$ is the area of land use type $i$, $VC_i$ represents the value coefficient of the category $i$, and $VC_{if}$ represents the individual service value factor.

2.3.4. Cross Sensitivity Coefficient

The cross-sensitivity coefficient indicates the extent to which the net conversion between two different land types contributes or inhibits the value of ecosystem services. The calculation formula is as follows [49]:

$$P_{cicskt} = \frac{(V_{ck} - V_{ct})\Delta S_{kt}}{\Delta P_{ESV}} \tag{18}$$

where $P_{cicskt}$ represents the cross-sensitivity coefficient for the transformation of category $k$ into category $t$. Its larger absolute value indicates that the $ESV$ is more sensitive to the net transformation of the two categories and otherwise less sensitive. $V_{ck}$ and $V_{ct}$ denote the equivalence factor of $ESV$ corresponding to $t$. $\Delta S_{kt}$ denotes the net conversion between category $k$ and category $t$. $\Delta P_{ESV}$ represents the amount of change in ESV during the study period.

**3. Results**
*3.1. Analysis of the Components of Land*
3.1.1. Analysis of Total Land Use Component Change

Figure 3 shows the components of total LULC in the Xiangjiang River Basin. In general, the total land use change in the watershed shows an increasing trend, with a sharp increase in the total change from 1990 to 2000, and the growth of displacement change is more obvious than both quantity and exchange change, with the shift change increasing sharply from 2000 to 2010. In terms of each time, the form of change in the first two periods is dominated by quantity change and in the last two periods by shift change.

3.1.2. Analysis of the Change Components of Different Categories

Figure 4 shows the components of change across the Xiangjiang River Basin at different times, with 'G' on the bar indicating a net increase in the category and 'L' indicating a net decrease in the category; the red dashed line represents the intensity of overall quantitative change; the black dashed line represents the sum of the intensity of overall quantity and exchange change. Figure 4 shows that the overall intensity of change in volume for 1980–1990, 1990–2000, 2000–2010, and 2010–2020 is 55%, 52%, 52%, and 32%, respectively,

and their overall intensity of change in exchange is 29%, 41%, 38%, and 65%, respectively. During the period 1980–1990 (Figure 4a), there was a net decrease in cultivated, forest, and bare land and a net increase in the grass, water, and built-up in the Xiangjiang River Basin, with cultivated, water, built-up, and bare land dominated by quantity changes and forest and grass dominated by exchange and shift changes. Between 1990 and 2000 (Figure 4b), cultivated, forest, and grass showed a net decrease; water, built-up and bare land realized a net increase, with cultivated, water, and built-up dominated by quantitative changes; the land types dominated by exchange changes were forest, grass land, and bare land, respectively. Between 2000 and 2010 (Figure 4c), there was a net decrease in cultivated and grass; a net increase in the forest, water, built-up, and bare land, with cultivated, grass, built-up, and bare land being predominantly quantitative; and forest and water being predominantly exchanged changes. From 2010 to 2020 (Figure 4d), there is a net decrease in the realization of cultivated, forest, grass, and bare land and a net increase in water and built-up. Of these, the volume change in all categories except for bare land gradually diminished, with exchange change dominating for cultivated, forest, grass, and water, and although build-up and bare land remain dominated by volume change, the proportion of built-up land is relatively weaker, and the proportion of bare land has increased, accounting for 71% and 63% of the overall category, respectively.

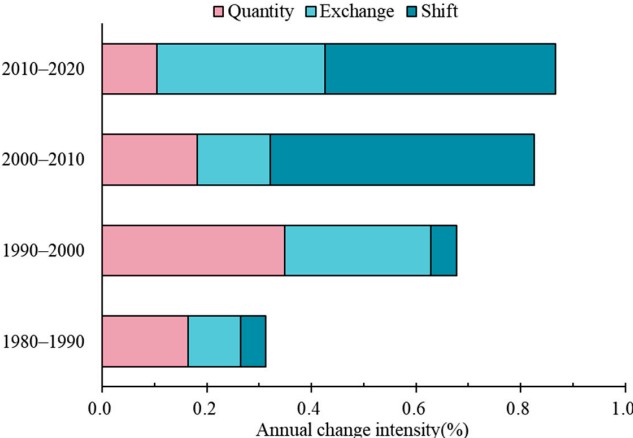

**Figure 3.** Change in the total composition at different time intervals.

*3.2. Intensity Analysis of LULC*

3.2.1. Interval Level

Figure 5 shows the results of the analysis of the interval level. The red dashed line indicates the intensity of equilibrium change in LULC from 1980 to 2020. If there are bars above the uniform line, the interval is fast; otherwise, it is slow. As can be seen in Figure 5, the intensity of land use change in the Xiangjiang River Basin over the past 40 years shows a marked increase, with the first interval showing a relatively slow change and the last three intervals realizing a rapid change. Meanwhile, Figure 5 also shows that, in Period 4, the land in the Xiangjiang River Basin is more influenced by socioeconomic and human activities, resulting in an approximately three-fold increase in the rate of change compared to Period 1.

3.2.2. Category Level

Figure 6 shows the changes of the different categories at four intervals, where the red dashed line indicates the average annual intensity of change for each interval. The right of the uniform line indicates that the annual gain or loss of the category of land is active. The left of the uniform line indicates that the annual gain or loss of the category of land is dormant. As can be seen in Figure 6, during the period 1980–1990, the intensity of decrease in cultivated, grass, water, and bare land was active, while the intensity of increase in the grass, water, built-up, and bare land was active. During the period 1990–2000, the intensity

of decrease in cultivated, grass, and bare land was active, and the intensity of increase in the grass, water, built-up, and bare land increased significantly. In the period 2000–2010, the intensity of the decrease in cultivated, grass, water, built-up, and bare land was active, and the intensity of the increase in the grass, water, built-up, and bare land was active, with bare land increasing significantly. From 2010 to 2020, the decrease of cultivated, grass, water, built-up, and bare land remains active, the increase of cultivated, grass, water, and built-up remains relatively active, and the increased intensity of bare land changes to a stable state. The forest has remained relatively stable throughout the study period.

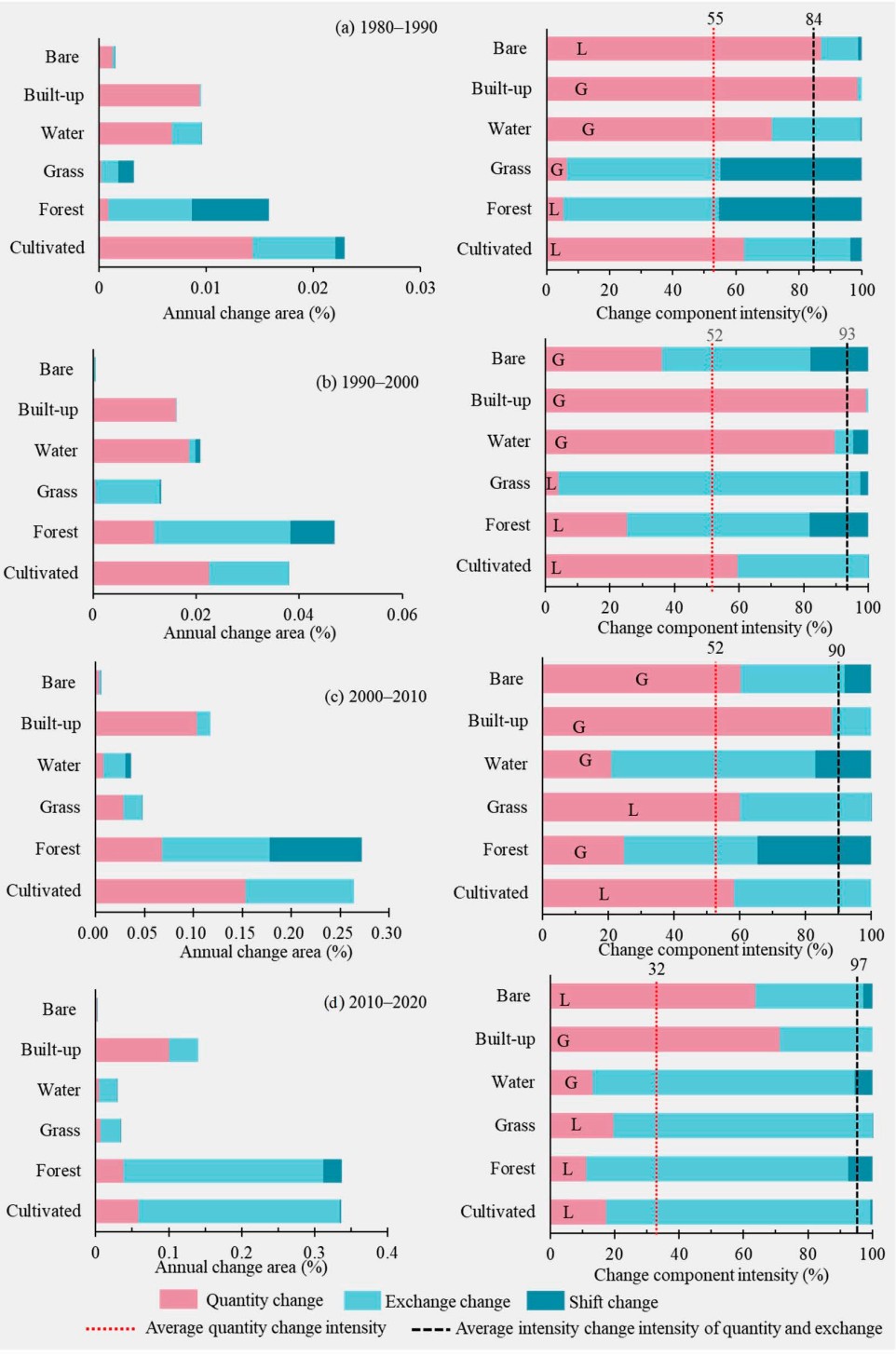

**Figure 4.** Change component analysis of the category at different periods.

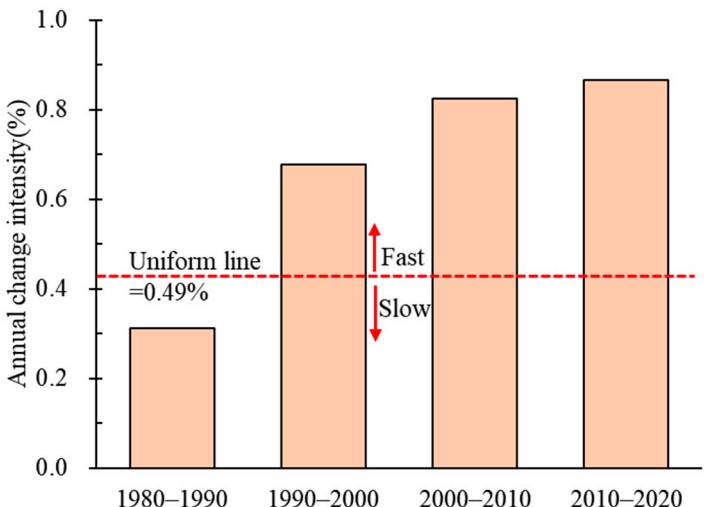

**Figure 5.** Total intensity of the change and change component intensity during 1980~2020.

### 3.2.3. Transition Level

Figure 7 shows the transfer of other categories to built-up land. The left side shows the transfer size in each time interval, and the right side shows the transfer intensity, in which the dashed blue line shows the intensity of uniform change. If it exceeds the intensity of uniform change, then this category tends to transition to construction land; otherwise, it means that this category avoids the transition to built-up land. Figure 7 shows that the largest contributors to the increase of built-up land are cultivated and forest land, in which the main source of the increase of built-up land area is cultivated land, with water and bare land also showing a trend of transformation to built-up land between 1990 and 2010. In addition, among the five categories converted to built-up, the intensity of change in grass and forest land was lower than the average intensity of change over four-time intervals, demonstrating that the conversion process of these two categories to built-up showed a stable pattern of avoided transformation.

### *3.3. Analysis of Ecosystem Service Value*

#### 3.3.1. Temporal Changes in Ecosystem Service Values

The ESV of different land types in Xiangjiang River Basin from 1980 to 2020 was calculated by the equivalent factor method, and the calculation results are shown in Table 3. Land use change in the Xiangjiang River Basin caused the change of ESV. According to the analysis, from the total amount of ESV, the ESV in the basin increased first and then decreased from 1980 to 2020, the increase from 2000 to 2010 was the largest, and in 2010, it reached the maximum value in the study area.

**Table 3.** ESV of different land use type in the Xiangjiang River Basin from 1980 to 2020.

| Land Use Type | Value ($10^8$/CNY) | | | | |
| --- | --- | --- | --- | --- | --- |
| | 1980 | 1990 | 2000 | 2010 | 2020 |
| Cultivated | 617.93 | 615.20 | 610.90 | 581.62 | 570.40 |
| Forest | 3712.84 | 3712.32 | 3705.22 | 3745.96 | 3723.19 |
| Grass | 56.33 | 56.37 | 56.28 | 50.59 | 49.27 |
| Water | 229.11 | 237.77 | 261.48 | 271.15 | 276.23 |
| Bare land | 0.10 | 0.09 | 0.09 | 0.13 | 0.11 |
| Total | 4616.31 | 4621.75 | 4633.96 | 4649.46 | 4619.21 |

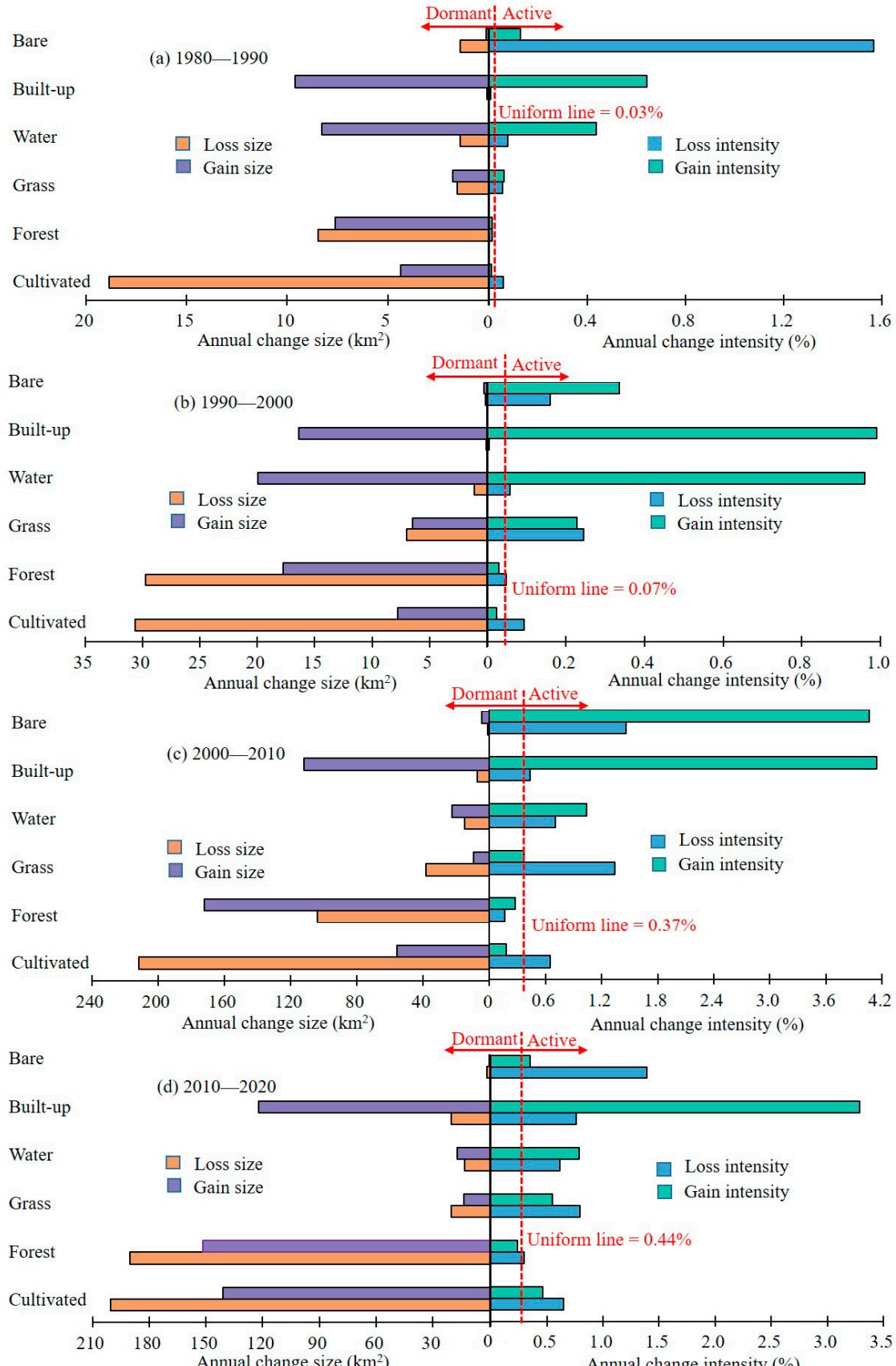

**Figure 6.** Annual size and intensity by category at four intervals.

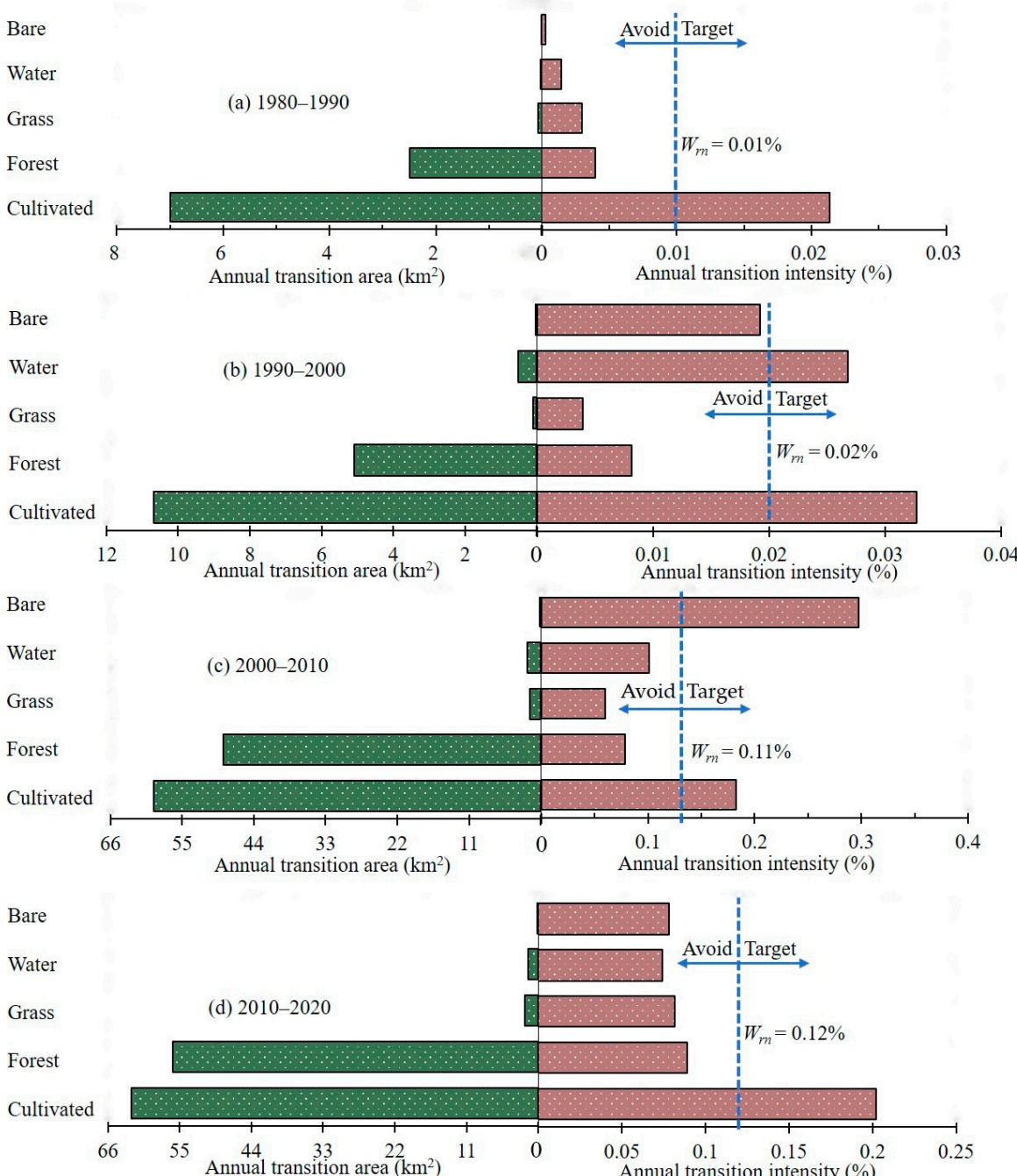

**Figure 7.** Annual transition size and intensity of the transition from other categories built up at four intervals: ((**a**) 1980–1990, (**b**) 1990–2000, (**c**) 2000–2010, and (**d**) 2010–2020).

Table 4 shows that hydrological regulation, gas regulation, and soil conservation are the top three individual ESVs. From 1980 to 2020, food production, gas regulation, climate regulation, soil conservation, and the maintenance of biodiversity in the Xiangjiang River Basin have generally decreased in each individual ESV, while the rest have generally increased, with the largest deficit being soil conservation The largest deficit was in soil conservation, with a loss of approximately RMB 1015 million, mainly due to the reduction in woodland areas over the period; the largest surplus was in hydrological regulation, with an increase of approximately RMB 1736 million, mainly due to the increase in water area over the period.

**Table 4.** Individual ESV of the Xiangjiang River Basin from 1980 to 2020.

| Ecosystem Service Functions | Value ($10^8$/CNY) | | | | |
|---|---|---|---|---|---|
| | **1980** | **1990** | **2000** | **2010** | **2020** |
| Food production | 109.29 | 108.91 | 108.31 | 102.38 | 102.23 |
| Raw material production | 272.46 | 7083.88 | 136.23 | 27.25 | 451.69 |
| Gas regulation | 645.67 | 645.40 | 643.94 | 647.72 | 643.11 |
| Climate regulation | 547.67 | 547.35 | 546.14 | 546.80 | 542.43 |
| Hydrological regulation | 705.12 | 708.65 | 717.73 | 724.69 | 722.48 |
| Waste disposal | 469.92 | 472.67 | 480.57 | 478.81 | 476.55 |
| Maintaining soil | 808.56 | 807.91 | 805.72 | 805.20 | 798.41 |
| Maintain biodiversity | 639.60 | 639.81 | 639.73 | 641.03 | 636.55 |
| Aesthetic Landscape | 240.78 | 241.58 | 243.44 | 246.65 | 245.78 |

3.3.2. Spatiotemporal Change of Ecosystem Service Value

In order to study the spatial distribution and variation of ESV in the Xiangjiang River Basin, this paper takes the grid scale as the research unit. By referring to the construction of relevant grids [50], a grid of 5 km × 5 km was formulated by comprehensively considering the size of the study area and the scale effect of the calculation results. Based on this, the ESV of each grid was calculated in this paper, and it was divided into five grades (Unit: billion CNY): lower [0–0.4], low [0.4–0.8], medium [0.8–1.2], high [1.2–1.6], and higher [1.6–3.2] by combining the natural segment point method and the level of ESV, and the spatial distribution pattern of ESV in the Xiangjiang River Basin in five periods was obtained (Figure 8). The ecosystem service value per unit area of the Xiangjiang River Basin was mainly of medium and high levels, and the ESV in the southeast side of the basin changed most obviously, and most of them were converted from low to high levels. In the Xiangjiang River Basin of ESV exist obvious east–west high–low space differentiation, namely the eastern ESV high-value area of the mainland types of forest land and cultivated land, mainly due to the good natural ecological background, these regions to balance its ESV loss brought by the construction land expansion, and the relatively low ESV regional urbanization process in the Midwest broke the balance of the ecosystem.

*3.4. Impacts of Land Use on Ecosystem Service Values*

The cross-sensitivity coefficients of net transition among different land use types in four periods in the Xiangjiang River Basin are shown in Figure 9. Since the net conversion area between the two land classes is bidirectional and symmetric, so is the cross-sensitivity. Therefore, this paper only shows the sensitivity coefficient of a single item to reveal the influence intensity of net conversion between different land classes. Figure 9 shows that ESV is sensitive to the net conversion between cultivated and other categories, where it is most sensitive to the net conversion between cultivated and forested land, with a cross-sensitivity coefficient greater than 1. From 1980 to 2020, the transformation between cultivated land and forestland was the net transfer from cultivated land to forestland, and the cross-sensitivity coefficient increased first and then decreased, which promoted the change of ESV in four periods. The net conversion between cultivated land and water area inhibited ESV change at first and then promoted it. The conversion between forest land and grassland showed a net transfer from forest land to grassland from 1980 to 1990, and the ESV increased. From 1990 to 2020, there was a net transfer from forestland to grassland, and ESV increased, which inhibited the change of ESV. The transformation between forestland and water area resulted in the reduction of forestland, and the cross-sensitivity coefficient was greater than 0, which promoted the change of ESV. The transition between forestland and unused was opposite to that between forestland and water area, and the sensitivity decreased, which showed the effectiveness of the implementation of the forest ecological protection policy. The net conversion between grassland and unused land and other land classes resulted in the decrease of grassland and woodland, and the sensitivity coefficient of grassland and woodland decreased continuously.

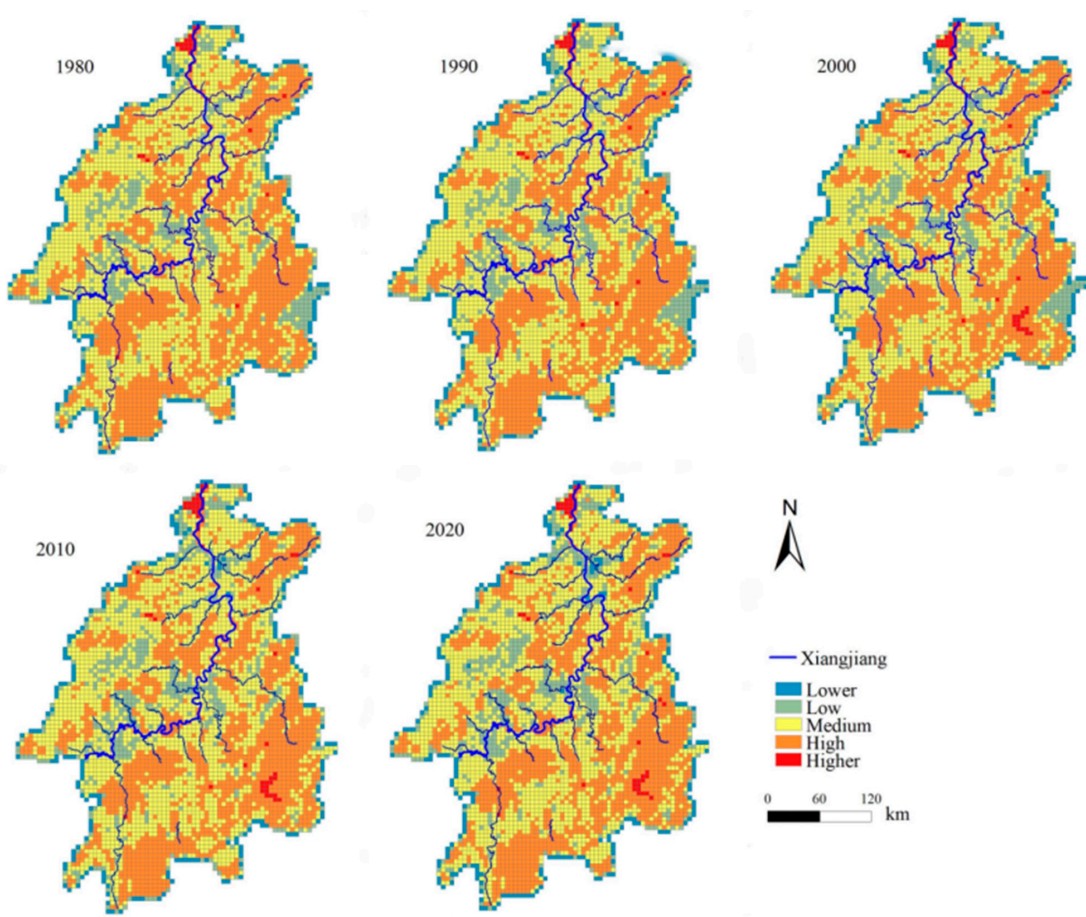

**Figure 8.** Spatial variation of ESV in the Xiangjiang River Basin from 1980 to 2020.

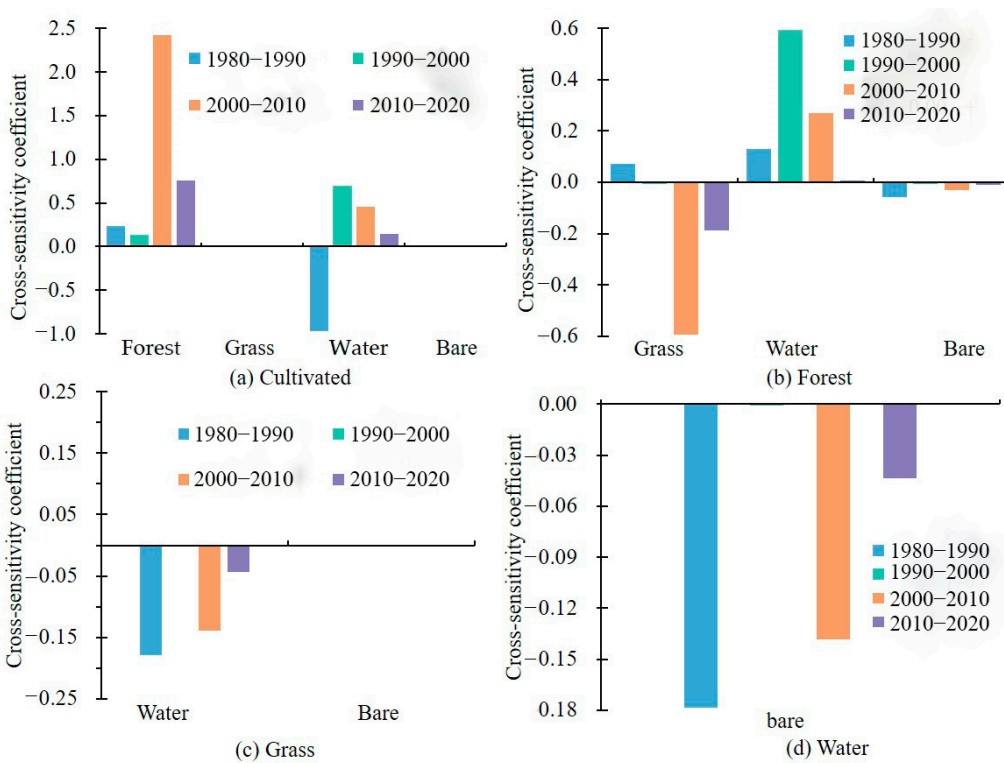

**Figure 9.** Cross-sensitivity coefficient of ecosystem services in Xiangjiang River.

## 4. Discussion

### 4.1. Pattern and Process of LULC

In the science of land change, the mode and process of LULC change are often discussed to analyze the characteristics of historical land change [51,52], but researchers often only observe the quantity change, and it is difficult to find the spatial instability of different land types. Thus, the composition change and Intensity Analysis methods are helpful because they reveal the pattern, magnitude, and intensity of the terrestrial transformation, which helps the reader to consider the possible causes of the apparent transformation. For example, Figure 3 shows that LULC changes in the Xiangjiang River Basin experienced a complex pattern from 1980 to 2020. In the four intervals, the first two intervals were dominated by quantity changes, and the last 20 years were dominated by shift changes, but there were different types of changes in different periods. Compared with the method that only considers the quantity change, the exchange and shift components of the changing components can be characterized as more terrestrial spatial transfer information [53,54]. Figure 4 shows that the quantity of forest and grass does not change significantly, but the proportion of exchange and displacement changes increases, which means that they may decrease in one area but increase in another area.

Through Intensity Analysis, the stability characteristics of land use change patterns in the study area can be identified, and these change patterns can be linked to the process, which is also helpful for researchers to better understand the dynamic mechanism behind these key patterns and processes and to link land use change patterns with socioeconomic change processes. Figure 6 shows that, during 1980–2020, the area of cultivated and forest decreased the most, the area of cultivated and grass increased the most, and the area of bare land increased and decreased the least. During 1980–2000, the area of cultivated and built-up land showed an increasing state. On the one hand, this can be explained by the large initial area of cultivated land and forest land in the Xiangjiang River Basin; on the other hand, the population scale and economic and social development in the Xiangjiang River Basin changed significantly from 1990 to 2000, with the population size increasing from 43 million in 1990 to 46 million in 2000. During this period, farmers' livelihoods mainly depended on agricultural production [55,56]. Since 2000, the living demand of farmers has been continuously improving, and more and more farmers choose to work in cities, and the phenomenon of abandoning farmland gradually appears. In addition, since 2001, Hunan Province has officially implemented the project of "returning farmland to the forest", resulting in the growth of cultivated and forest area from 2000 to 2020.

### 4.2. The Relationship between LULC and ESV

The change of landscape pattern caused by the intensity of LULC affects the material circulation and energy flow of the ecosystem, resulting in endogenous watershed changes in the value of state system services [57,58]. Our results show that the built-up and water area have increased, and the area of cultivated land and forest land is decreasing. This was similar to the conclusion of Han et al. and Chen et al. in the Xiangjiang River Basin [47,48]. Compared with related studies that rely on traditional land use analysis methods [59,60], Intensity Analysis methods can analyze the land use information behind ESV changes from multiple perspectives [46]. Our study found that the ESV growth from 1980 to 2010 was mainly caused by the ESV growth of water, and the contribution rate of ESV growth in water depended on the change of area. From 2010 to 2020, the total ESV of the river basin was reduced, and the comparative analysis was mainly the reduction of ESV provided by forest land and arable land. We analyzed layer by layer with the Intensity Analysis and component analysis. First, it can be seen from the change composition that the area of water from 1980 to 2010 showed a net increase, and it was mainly based on quantitative changes and exchange changes, indicating that the increase in water area during this period led to an increase in ESV (Figure 4). Second, from the perspective of the category level (Figure 6), the annual increase intensity of water gradually increases, the increased intensity is greater than the decreased intensity, and the increased area is also greater than the reduced area, so

that the ESV provided by the water during this time increases. However, during the period 2010–2020, the reduction of ESV at the category level showed that the intensity of increase in cultivated land and forest was greater than the intensity of reduction, and the area of cultivated land and forest decreased by more than in the previous time interval. Third, from the perspective of transfer level (Figure 7), the intensity of cultivated land and forest land transferred to construction land is greater than that of other land types, and secondly, the area of cultivated land and forest land transferred to it has also increased, but the positive environmental benefits of construction land are smaller than the negative benefits.

In addition, our study found that the value of ecosystem services in the Xiangjiang River Basin reached the greatest in 2010, mainly because Hunan Province was approved as a pilot area for supporting reform of the construction of a "two-type society" from 2000 to 2010, and a series of construction activities of "returning land to lakes" and "returning farmland to the forest" was carried out in the research area, and the areas of water and forest land also increased. From the perspective of the spatial distribution of ESV (Figure 8), the spatial pattern of ecosystem service value in the Xiangjiang River Basin remained stable during the study period, and the high-value area was mainly distributed in the southeast forest and its surrounding areas. Therefore, the Xiangjiang River Basin area should take land and spatial planning as the starting point, reasonably divide ecological function areas, and build hierarchical and differentiated spatial planning strategies. For high-value areas, forest land construction and water ecosystem restoration should be carried out. For areas with the low value of ecosystem services, attention should be paid to their ecological construction and ecological restoration.

## 5. Conclusions

(1) Over the past four decades, land use in the Xiangjiang River Basin has undergone dramatic changes, the intensity of which has shown a continuous increase, mainly in the form of quantity changes and shift changes. The increase in built-up land and bare land and the decrease in cultivated land are stable and active, and the loss of forest land is large.

(2) In the past 40 years, the ESV of the Xiangjiang River Basin has significantly changed in time, first increasing and then decreasing, and the spatial changes are most obvious in the middle and southeast of the basin.

(3) The value provided by land use types was in the order of forest land > cultivated land > water area > grassland > unused land, and the cross-sensitivity coefficient reflected that the net conversion between cultivated land and forest land and water area had a promoting effect on ESV.

**Author Contributions:** Conceptualization, B.Q.; methodology, Z.Z.; supervision, B.Q.; funding acquisition, B.Q.; formal analysis, B.Q. and Z.Z.; investigation, Z.Z. and Z.D.; and writing—original draft, Z.Z. All authors have read and agreed to the published version of the manuscript.

**Funding:** This work was supported by the key project of the Social Science Foundation of Hengyang under grant number 2021B(I)004 and the Open Foundation of Hengyang Base of International Centre on Space Technologies for Natural and Cultural Heritage with the auspices of UNESCO under grant number 2021HSKFJJ029.

**Acknowledgments:** Thanks to Robert Gilmore Pontius Ir. From Clark University for creating the Intensity Analysis method of land use change and "PontiusMatrix42A.xlsx".

**Conflicts of Interest:** The authors declare no conflict of interest.

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
