# Peer review of "Effects of Land Use Changes on Ecosystem Service Value in Xiangjiang River Basin, China"

_sustainability, doi:10.3390/su15032492_

Round 1
Reviewer 1 Report
The effort of LULC changes and its impact on ESV is Tanzania is interesting and will bring a significant contribution in this field.
Besides this manuscript needs serious improvement.
It is a good idea to add in 2.1. subsection some information about economic activity in this region. Is it industrial or agricultural at present time? It is important to understand at the beginning the importance of such research according to main drivers in this region
It is a good idea to add LULC maps to every time step to understand spatio-temporal distribution of different classes
Please add any information about software, packages, libraries which you are used in this research.
I wish that my comment would be helpful in improving the quality of this research.
Thank you.
Author Response
Dear editor Irene Zhangand dear reviewers:
Thank you for your letter and the reviewers' comments on our manuscript entitled "Effects of Land Use Changes on Ecosystem Service Value in Xiangjiang River Basin, China" (sustainability-2157317). The comments were valuable and very helpful. We have read the comments carefully and have made corrections. In accordance with the instructions in your letter, we have now uploaded the revised manuscript file. Please check the attachment for your specific response comments. Our manuscript has been marked up using the 'track changes' feature in Word 2016.
We would like to thank you for allowing us to resubmit the revised manuscript and we highly appreciate your time and consideration.
Manuscript ID: sustainability-2157317
Title: Effects of Land Use Changes on Ecosystem Service Value in Xiangjiang River Basin
Your sincerely,
Authors

Reviewer 2 Report
General comments: The dramatic changes in land use have become a major factor in the erosion and damage of ecosystem service values in the context of rapid urbanisation. Exploring the impact of land use changes in watersheds on the value of ecosystem services is of great practical significance in identifying the state of the regional ecological environment and optimising the spatial pattern of the national territory.
Based on five periods of land use data, this paper uses quantitative intensity analysis and component change analysis to make a detailed analysis of land use change over multiple periods in the Xiangjiang River Basin and to explore the spatial and temporal changes in the value of ecological services and the impact of land use change on the value of ecosystem services. This is indeed an interesting study. The findings and analysis can clearly present the processes and characteristics of land use change in the Xiangjiang River Basin as well as explain the impact of land use change on ESV from multiple viewpoints, and support the concepts well. Some important findings can be pointed out in this paper, such as the trend of enhanced land use intensity change in the Xiangjiang River Basin and the net conversion between cropland and forest land contributing the most to ESV.
Major Comments:
1. In Figure 1, is the boundary range of Xiangjiang River Basin accurate? Why does the boundary of the basin coincide with the boundary of Hunan Province? If there are existing relevant research papers that carry out research on this catchment scope, the references should be added.
2. Equation (1) - (8), Please add a note for each mathematical symbol, if it appears in Table 1, it needs to be written in the text.
3. L192—Please correct the format of the references.
4. L239-322, Why choose a 5km*5km grid? What is the difference between the commonly used 500 m × 500 m, 1 km × 1 km and 3 km × 3 km grid cells?
5. What is the range of the "lower, low, medium, high, higher" division in Figure 7? should be given explicitly in the text or in the figures.
6. In Fig.1. Please keep the scales the same size.
7. In Fig.7. Adjust the distance between "Avoid" and "Target" and the blue dotted line and keep the spacing consistent.
Author Response

(The authors gave the same response as above.)

Reviewer 3 Report
Zhou et. al systematically evaluated the land use changes and ecosystem service value in Xiangjiang river Basin, a critical area of agricultural modernization in Hunan Province on a long-term scale using change composition and intensity analysis. The result provides detailed information and is valuable resources for both scientific research and policy make. The presented data and method are convincing with detailed explanation. However, there’re some minor limitations (as outlined below) that limit the enthusiasm for publishing its current form.
1. In general, the figure legends are too simple to make it clear to the reader. Could the author please provide more description in figure legends?
2. In line 174, should the number of equations “3 and 4” change to “11 and 12”?
3. In Table 4, the “Raw material production” of 2020 is 0. Is this because of missing data or this is the real situation. Could the author make this clearer?
Author Response

(The authors gave the same response as above.)

Reviewer 4 Report
The paper is very interesting, analysing the trend of land use in specific area.
I am not a specialist of land use change and the evolution of ecosystem service (ESV) but, in my opinion, it could be very interesting to integrate part of such complex analysis in tools for the evaluation of ESV and standard environmental impact (LCA- land use). In the discussion, this fact could be addressed and developed.
Author Response

(The authors gave the same response as above.)
